# Mediating Role of PERMA Wellbeing in the Relationship between Insomnia and Psychological Distress among Nursing College Students

**DOI:** 10.3390/bs13090764

**Published:** 2023-09-14

**Authors:** Qian Sun, Xiangyu Zhao, Yiming Gao, Di Zhao, Meiling Qi

**Affiliations:** 1School of Physical Education, Shandong University, Jinan 250012, China; sunqian66@sdu.edu.cn; 2School of Nursing and Rehabilitation, Cheeloo College of Medicine, Shandong University, Jinan 250012, China; zhaoxiangyu0717@163.com (X.Z.); gaoyiming0401@163.com (Y.G.); 15069066196@163.com (D.Z.)

**Keywords:** insomnia, psychological distress, PERMA, college students

## Abstract

Background: Psychological distress is an important mental health problem in college students. Insomnia may be a major factor contributing to psychological distress. This study aimed to explore the indirect relationship between insomnia and psychological distress through the five PERMA wellbeing variables (i.e., positive emotions, engagement, relationships, meaning in life, and achievement) among nursing college students. Methods: A cross-sectional study was conducted in China using an online survey design. Mediation analyses were examined using the PROCESS macro version 4.1 for SPSS 27.0. A total of 1741 nursing college students completed the online survey. Results: Insomnia was positively associated with psychological distress (*p* < 0.01, *r* = 0.673), while negative associations were detected between PERMA wellbeing variables and insomnia (*p* < 0.01, *r* range = −0.176 and −0.272), as well as psychological distress (*p* < 0.01, *r* range = −0.196 and −0.386). The association between insomnia and psychological distress was partially mediated by the participants’ positive emotions (indirect effect = 0.137, SE = 0.024, 95% CI boot = [0.094, 0.188]), engagement (indirect effect = −0.033, SE = 0.010, 95% CI boot = [−0.054, −0.017]), and meaning in life (indirect effect = 0.027, SE = 0.014, 95% CI boot = [0.001, 0.055]) but not their relationships or achievement of the PERMA wellbeing variables. Conclusions: The findings of this study suggest that the PERMA wellbeing variables, especially positive emotions, engagement, and meaning in life, could be potential mechanisms by which insomnia is associated with psychological distress. The mediating roles of PERMA wellbeing variables between insomnia and psychological distress could be incorporated into the health management of university administrations to promote the health and wellbeing of nursing college students.

## 1. Introduction

Psychological distress has become an important public health problem, with the prevalence rate of psychological distress (e.g., stress, depression, and burnout) being over 50% among college students [1]. Higher levels of psychological distress have a potential role in negatively influencing college students’ academic performance, academic self-perception, physical functioning, and overall wellbeing, and even decreasing their quality of life [2,3]. It is, therefore, important to explore the relevant factors associated with college students’ psychological distress and consider these features in the design of an effective strategy to decrease their levels of psychological distress. College students may experience greater severity levels of stress, anxiety, and sadness because of the changes in their surroundings and lack of interpersonal relationships after being far away from home [4,5,6]. Recent studies found that more than half of university students reported longer time periods spent on mobile phones and Internet gaming during the COVID-19 pandemic [7,8,9], which have also been suggested to be potential risk factors for psychological distress. Other potential factors such as a disadvantaged family background, excessive course load, and insufficient activity also have an impact on psychological distress [10,11,12].

Although several potential reasons for psychological distress have been explained, recent studies revealed that sleep quality was a potential factor contributing to psychological distress and poor psychological health among college students [13]. Becerra [13] indicated that most college students reported sleep periods of less than 8 h per night. Sleep disorders are highly prevalent among nursing students [14]. A systematic review conducted by Scott [15] indicated that sleep was causally related to the experience of mental health difficulties. Another study also demonstrated the positive associations between insomnia and psychological distress in Japanese nursing students [16]. Interestingly, the existing evidence suggested that there was a bidirectional association between insomnia and psychological distress [17], whereas one study pointed to sleep as a stronger influence on psychological distress than the reverse pathway among college students [18]. These findings support the fact that insomnia may be an essential factor of psychological distress to be taken into consideration among college students. Furthermore, one study found that loneliness may be a potential mediator of insomnia and psychological distress [18]. However, other wellbeing mechanisms by which insomnia might be associated with psychological distress among nursing college students still need to be identified.

The PERMA variables designed by Seligman [19] explore individuals’ wellbeing in five elements (i.e., positive emotions, engagement, relationships, meaning in life, and achievement). Each element of PERMA could help to improve individuals’ health and wellbeing. A growing body of empirical evidence indicated the negative associations between insomnia and positive emotions in college students [20,21]. Mou [22] also indicated that poor sleep quality contributed to low academic engagement among Chinese university students. In addition, social connections with family and friends facilitate the positive relationship between insomnia and psychological distress in college students [18]. Furthermore, inadequate sleep causes impaired cognitive function and is associated with worse academic achievement in higher education students, whereby insomnia was a predictor of lower grades (last exam, average in current academic year) [23]. Insomnia resulting in burnout has also been found to have a negative correlation with students’ academic ability [2]. These study findings suggest that insomnia may be negatively associated with college students’ PERMA wellbeing variables.

A German longitudinal study demonstrated that positive mental health partially mediated the association between addictive Internet use and depressiveness [20]. Other positive emotions, such as regulatory emotion self-efficacy and interpersonal adaptability, are also negatively correlated with psychological distress among medical students [24]. Greater engagement in meaningful activities was also associated with better emotional functions and decreased mental disorders [25]. A good relationship with others was also linked to great resilience and in turn decreased levels of psychological distress in university students [26]. Meaning in life refers to an innate drive to find significance in individuals’ lives, which has been demonstrated to be a protective factor that is closely related to decreased mental disorders (e.g., stress and anxiety) during the outbreak of major infectious diseases [27,28]. In addition, students’ higher academic achievement levels have been proven to be a mediator between the school ethos and mental health, and it was negatively associated with college students’ psychological distress [29].

In summary, the PERMA wellbeing variables may be a potential mechanism to adjust the relationship between insomnia and psychological distress. Importantly, one study explored the role of wellbeing in the prediction of psychological distress and revealed that PERMA wellbeing showed a higher prediction of depression and acted as a mediator in depression and stress [30]. However, limited studies have examined the potential mediating role of PERMA wellbeing in insomnia and psychological distress among college students. Hence, this study aimed to estimate the indirect relationship between insomnia and psychological distress through the PERMA wellbeing variables among nursing college students in China. This study hypothesized the following: (a) there is an indirect relationship between insomnia and psychological distress; (b) PERMA wellbeing variables (i.e., positive emotions, engagement, relationships, meaning in life, and achievement) play mediating roles between insomnia and psychological distress.

## 2. Materials and Methods

### 2.1. Study Design and Participants

A cross-sectional study using an online survey design was performed in this study. Convenience sampling was used in this study. The data collection took place between 10 October and 30 December 2021. Ethics approval for this study was received from the Human Research Ethics Committee of the affiliated institute (2021-R-165). After gaining ethics approval, emails seeking participants were sent to a large medical college in China by the College Academic Affairs Office. The email specified that this online survey was open to nursing students. Potential participants were directed to click on an online survey link containing the study purpose, electronic consent information, inclusion and exclusion criteria, and research questionnaire. Importantly, the completion and submission of the online survey implied consent to participate. This was declared to respondents at the commencement of the survey. This study involved a secondary analysis of cross-sectional data, with the original study using a latent profile analysis to identify the classes of wellbeing based on the five PERMA wellbeing variables [31].

### 2.2. Demographic Details

The participants’ demographic information (i.e., age, gender, and year of study) was collected in this study. The mobile phone use times per day during workdays and weekends of potential participants were also included in the data collection.

### 2.3. PERMA Wellbeing (Mediator)

The Chinese version of PERMA-Profiler was used to evaluate individuals’ multidimensional wellbeing [32]. This study used the 15 items of the PERMA-Profiler to assess the positive emotions, engagement, relationships, meaning, and achievement of participants’ PERMA wellbeing (i.e., five PERMA elements). The participants chose their feelings or behaviors on a Likert-type scale ranging from 0 to 10 (0 = not at all, 10 = completely; or 0 = never, 10 = always; or 0 = terrible, 10 = excellent). Of the 15 items, three items assessed one PERMA element. The score for each PERMA element was obtained from an average score across the three items. Higher scores indicated greater wellbeing. This tool demonstrated good internal consistency among Chinese nursing college students in this study (Cronbach’s α = 0.933). The PERMA wellbeing scale also showed high reliability, with Cronbach’s α being over 0.73 for each element among undergraduate students in one previous study [33].

### 2.4. Psychological Distress (Outcome Variable)

The Chinese version of the 10-item Kessler Psychological Distress Scale (K10) was used to assess the participants’ psychological distress in the past month [34]. The K10 is a self-reported questionnaire containing ten questions with scores ranging from 1 to 5 to assess participants’ frequency of non-specific psychological distress across the past month based on questions related to symptoms of anxiety and depression. Participants choose from the following options: 1 = almost never, 2 = sometimes, 3 = fairly often, 4 = very often, 5 = all the time. The total scores were obtained by summing all 10 items, with total scores of 10–50. Higher scores indicate higher levels of psychological distress, and a score of 22 or greater indicates a high level of psychological distress. The K10 scale is a valid instrument with acceptable internal consistency, with Cronbach’s α being over 0.954 in this study and Cronbach’s α being over 0.84 among adults over 18 years old in one previous publication [34].

### 2.5. Insomnia Symptoms (Predictor)

The Chinese version of the Insomnia Severity Index (ISI) was used to assess the participants’ symptoms of insomnia [35]. The ISI is a self-reported questionnaire containing seven questions with scores ranging from 0 to 4 to assess the participants’ degree of insomnia during the past week. The total scores were obtained by summing all seven items, with total scores of 0–28. High scores indicate a higher degree of insomnia. A score of seven or less reflects no insomnia, with mild insomnia scores ranging from 8 to 14, moderate insomnia scores ranging from 15 to 21, and high insomnia scores ranging from 22 to 28. High reliability (Cronbach’s α = 0.88) for the ISI was found for nursing college students in this study. The ISI also had high reliability for adults over 18 years old (Cronbach’s α = 0.91) in one previous publication [35].

### 2.6. Statistical Analysis

A data analysis was conducted using the Statistical Package for the Social Sciences (SPSS) version 27.0. The descriptive statistics were calculated using frequencies (i.e., percentages) for categorical variables and means and standard deviations for continuous variables. Differences in psychological distress based on demographic characteristics were compared using an independent *t*-test (i.e., gender) and Kruskal–Wallis H test (i.e., year of study). Furthermore, the associations between insomnia, PERMA wellbeing variables, psychological distress, and other participants’ characteristics (i.e., age and mobile use times) were examined using a Pearson correlation analysis. Significant demographic and basic covariates were controlled in the subsequent analyses.

The mediating effect was examined using the PROCESS macro version 4.1 for SPSS, model 4. The bootstrapping was set to 5000 samples to provide robust estimates of the 95% confidence intervals (CIs) of the standardized effects. The absence of zero from a 95% CI indicated a significant direct or indirect effect. This study examined whether the five PERMA wellbeing variables mediated the association between insomnia and psychological distress separately. A model was constructed with insomnia as the predictor (X); psychological distress as the outcome (Y); and positive emotions, engagement, relationships, meaning in life, and achievement as the mediators (M). The covariates included the year of study and average mobile phone use time per day on workdays and weekends. The *p* values were two-tailed and the statistical significance level was set at *p* < 0.05

## 3. Results

### 3.1. Associations between Participant Characteristics and Psychological Distress

A total of 1741 nursing college students completed this study. Table 1 presents the associations between the participants’ characteristics and psychological distress. There were significant differences in the psychological distress scores between the years of study (*H* = 21.641, *p* < 0.001). The participants’ psychological distress scores also had positively significant associations with the average mobile use times per day on workdays (*p* = 0.01, *r* = 0.062) and on weekends (*p* < 0.002, *r* = 0.072).

### 3.2. Associations between PERMA, Insomnia, and Psychological Distress

As described in Table 2, there were significant associations between the scores for the five PERMA wellbeing variables, insomnia, and psychological distress. There was a positively significant association between the scores for insomnia and psychological distress (*p* < 0.01, *r* = 0.673). Participants who experienced greater insomnia reported higher levels of psychological distress. In addition, the participants’ insomnia scores were found to have negative associations with scores of positive emotions, engagement, relationships, meaning in life, and achievement (*p* < 0.01, *r* range = −0.176 and −0.272), indicating that the participants reported greater wellbeing when experiencing lower levels of insomnia. Similarly, significantly negative associations between psychological distress scores and PERMA scores (*p* < 0.01, *r* range = −0.196 and −0.386) were found, whereby participants who reported greater wellbeing experienced lower levels of psychological distress.

### 3.3. Regression and Mediation Analysis

Table 3 presents the results of the regression analyses of the mediating effects of PERMA wellbeing variables after controlling the covariates (i.e., year of study and mobile use time). Insomnia was significantly associated with psychological distress (*β* = 0.667, *p* < 0.001) without mediators, positive emotions (*β* = −0.270, *p* < 0.001), engagement (*β* = −0.123, *p* < 0.001), relationships (*β* = −0.249, *p* <0.001), meaning in life (*β* = −0.192, *p* < 0.001), and achievement (*β* = −0.168, *p* < 0.001). The direct effect of insomnia on psychological distress remained positive (*β* = 0.597, *p* < 0.001) with mediators. This study also found significant associations between positive emotions (*β* = −0.323, *p* < 0.001), engagement (*β* = 0.173, *p* < 0.001), and meaning in life (*β* = −0.088, *p* < 0.05) and psychological distress.

Importantly, the positive association between insomnia and psychological distress was partially mediated by the participants’ positive emotions (indirect effect = 0.137, SE = 0.024, 95% CI boot = [0.094, 0.188]), engagement (indirect effect = −0.033, SE = 0.010, 95% CI boot = [−0.054, −0.017]), and meaning in life (indirect effect = 0.027, SE = 0.014, 95% CI boot = [0.001, 0.055]) (see Table 4). The model is presented in Figure 1.

## 4. Discussion

This is the first known study to examine the mediating role of PERMA wellbeing variables in insomnia and psychological distress among nursing college students in China. We examined the direct and indirect relationships between insomnia and psychological distress, suggesting that insomnia is likely a unique and potential risk factor contributing to psychological distress among college students. Indeed, existing studies had found a bidirectional association between insomnia and psychological distress [17,18]. Furthermore, one previous study indicated the more important role of insomnia in psychological distress [18]. Taken together, these findings support the premise of using effective strategies to improve college students’ sleep quality in order to decrease their psychological distress and in turn promote their wellbeing.

It was hypothesized that each of the five PERMA wellbeing variables would mediate the positive association between insomnia and psychological distress among college students. However, this study did not reveal the significant mediating role of relationships and achievement in insomnia and psychological distress, which is not consistent with the results of at least four studies [18,23,26,29], demonstrating strong associations between insomnia, relationships, academic achievement, and psychological distress. The different results might be due to the choice of outcome measurements and their sensitivity to change in the study demographic. Philbrook and Macdonald-Gagnon [18] used the loneliness measurements to present the relationships with families and friends. In addition, the academic achievement of the students was assessed based on self-reported details regarding their grades and individual student-reported marks from different subjects in previous studies [23,29]. Further studies are needed to better verify the mediating effects of relationships and achievement on the association between insomnia and psychological distress.

In line with the study hypothesis, positive emotions and meaning in life, as part of PERMA wellbeing, are potentially mediating mechanisms linking insomnia and psychological distress among nursing college students. Compared with one study that found a moderate factor of PERMA in the relationships of COVID-19, depression, and stress [30], the current study used the five elements of PERMA wellbeing as predictors, whereas Carreno [30] used the whole wellbeing score as the predictor. However, this study’s findings further showed that some of the PERMA wellbeing variables also mediated the relationship between insomnia and psychological distress, with positive emotions and meaning in life being negative predictors of psychological distress. These indirect influence findings also indicated that insomnia could pose a negative influence on college students’ positive emotions and meaning in life. A lower level of insomnia may also increase the positive emotions and participants’ feeling of life meaning, in turn promoting the participants’ psychological wellbeing [25]. Hence, this study better explains the mediating role of positive emotions and meaning in life of the PERMA wellbeing in the relationship between insomnia and psychological distress.

This study also found that engagement partially mediated the relationships of insomnia and psychological distress. This indirect influence finding indicated that insomnia could pose a negative influence on college students’ engagement, which is consistent with one previous study suggesting that poor sleep quality could negatively influence students’ cognitive dysfunction, thereby reducing their motivation for academic engagement [13]. Interestingly, this study explored a positive association between engagement and psychological distress, which was not inconsistent with the previous study. Greater engagement in meaningful activities was also associated with better emotional functions and decreased mental disorders [25]. The COVID-19 pandemic significantly impacted college education in these years, where most of the class courses were shifted to online learning via electronic devices (e.g., mobile phones) [7]. A high screen exposure time due to online learning was found to be linked with increased levels of stress and headache [36], and students found that online learning affected their communication efficacy, which in turn caused increased psychological distress [37]. This study also found that nursing college students reported more than 5 h of mobile phone usage during workdays and weekends, and long mobile phone use times could pose a potential risk for psychological distress. Thus, engagement in online courses may harm college students’ psychological distress. Further research is needed to confirm the relationship between engagement and psychological distress and the mediating role of engagement in insomnia and psychological distress among college students.

The long mobile use times in this study are consistent with one recent study where more than 70% of university students used mobile phones for over 5 h per day [8]. Although studies found that long mobile use times may be also a potential risk factor contributing to poor sleep quality and low academic achievement of medical students [9,17], this finding suggests that effective interventions may need to reduce the mobile use times of nursing college students and in turn reduce their symptoms of psychological distress. However, it was difficult to determine the role of mobile phone use in insomnia in this study, as this study lacked an examination of the association between mobile phone use times and insomnia. It is necessary and important for future studies to explore the associations between mobile phone use times and insomnia in college students.

### 4.1. Practical Implications

These findings can be used in the study of nursing students’ health promotion within colleges. Nursing college administration workers play a vital role in managing nursing students’ wellbeing and performance in college settings. In particular, this study informs nursing college administration workers about psychological distress and insomnia among college nursing students so that they can consider strategies to support nursing students to effectively manage their situation. Furthermore, this study’s findings also provide preliminary evidence for the mediating role of positive emotions, engagement, and meaning in insomnia and psychological distress to the existing literature. Further researchers should develop educational or mental health interventions to reduce psychological distress and insomnia in college nursing students. Considerations should be given to the potential factors of PERMA wellbeing variables (e.g., positive emotion, engagement, and meaning) in the development of educational and mental health interventions to promote the health and wellbeing of nursing college students. Importantly, this study and the previous literature [25,37] also suggest that determining college nursing students’ symptoms of positive emotion, meaning in life, and engagement allows the early detection of psychological distress in this population.

### 4.2. Study Limitations

A number of limitations should be considered in future studies to explore the mediating role of PERMA wellbeing variables in insomnia and psychological distress. The cross-sectional design may be a limitation of this study, as the cross-sectional study did not allow the testing of the temporal precedence of mediators or the causality between variables. Longitudinal designs are recommended in future studies to draw causal conclusions. Although there were no significant differences in gender in terms of psychological distress, the convenient sampling method used in one college may be another limitation that weakens the generalizability of the study results to other study settings. Additionally, the collection of insomnia, PERMA wellbeing, and psychological distress data was completed using a self-reported online survey. A positive response bias caused by self-reported measures might influence the mediating role of PERMA wellbeing variables in insomnia and psychological distress. A further limitation of this study is the missing collection of tobacco and alcohol use information in the participants’ demographic details, as existing researchers found that the use of tobacco and alcohol may have a potentially negative influence on college students’ psychological distress and public health engagement [38,39]. However, this study included nursing students and most of them were females without the use of tobacco and alcohol in China. thus, it may be better to consider these potential factors in future studies to investigate the psychological distress of college students.

## 5. Conclusions

This study confirmed the direct association between insomnia and psychological distress among nursing college students in China. This study also revealed that PERMA wellbeing variables (i.e., positive emotions, engagement, and meaning in life) have great mediating effects on the relationship between insomnia and psychological distress. Future research should consider these potential mediators in designing relevant interventions to reduce psychological distress among college students.

## Figures and Tables

**Figure 1 behavsci-13-00764-f001:**
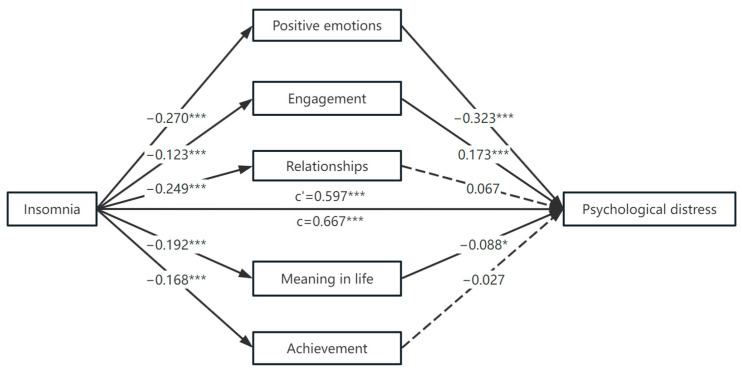
A mediation analysis of PERMA wellbeing variables’ effects on insomnia and psychological distress. Notes: * *p* < 0.05; *** *p* < 0.001; c: direct effect without mediators; c’: direct effect with mediators.

**Table 1 behavsci-13-00764-t001:** Associations between participant characteristics and psychological distress (n = 1741).

Variables	M ± SD	N (%)	Psychological Distress		*p*
			$\bar{x}$ ± SD	M (QL, QU)	
Age (years)	19.38 ± 1.02		18.67 ± 8.00		0.067 ^a^
Average mobile phone use time per day on workdays (h)	5.10 ± 2.34		18.67 ± 8.00		0.010 ^a^
Average mobile phone use time per day on workdays (h)	7.58 ± 2.73		18.67 ± 8.00		0.002 ^a^
Gender					0.692 ^b^
Male		375 (21.5)	18.52 ± 8.95		
Female		1366 (78.5)	18.72 ± 7.72		
Year of Study					<0.001 ^c^
First year		530 (30.4)		18 (13, 25)	
Second year		765 (43.9)		16 (11, 22)	
Third year		446 (25.6)		18 (11, 26)	

Note: ^a^ the value was calculated with the Pearson correlation analysis; ^b^ the value was calculated with the independent *t* test; ^c^ the value was calculated with the Kruskal-Wallis H test. QL = 25th percentile; QU = 75th percentile.

**Table 2 behavsci-13-00764-t002:** Associations between PERMA, insomnia, and psychological distress (n = 1741).

Variables	Test Scores	Correlation Coefficient (r)						
	M ± SD	1	2	3	4	5	6	7
1. Positive emotions	6.92 ± 2.05	1						
2. Engagement	6.28 ± 1.89	0.794 **	1					
3. Relationship	7.10 ± 2.05	0.869 **	0.732 **	1				
4. Meaning in life	6.63 ± 2.11	0.839 **	0.793 **	0.786 **	1			
5. Achievement	6.27 ± 2.00	0.806 **	0.779 **	0.769 **	0.898 **	1		
6. Insomnia	5.81 ± 5.07	−0.272 **	−0.122 **	−0.251 **	−0.198 **	−0.176 **	1	
7. Psychological distress	18.67 ± 8.00	−0.386 **	−0.196 **	−0.328 **	−0.310 **	−0.284 **	0.673 **	1

Note: ** *p* < 0.01.

**Table 3 behavsci-13-00764-t003:** Regression results.

Regression Model		Fitting Index		Significance of Coefficients	
Outcome Variables	Predictor Variables	R2	F	β	t
Psychological distress	Insomnia	0.459	294.513 ***	0.667	37.515 ***
Positive emotions	Insomnia	0.077	28.782 ***	−0.270	11.615 ***
Engagement	Insomnia	0.018	6.400 ***	−0.123	5.122 ***
Relationship	Insomnia	0.065	24.052 ***	−0.249	10.667 ***
Meaning in life	Insomnia	0.047	17.274 ***	−0.192	8.156 ***
Achievement	Insomnia	0.043	15.494 ***	−0.168	7.106 ***
Psychological distress	Insomnia	0.514	182.639 ***	0.597	33.537 ***
	Positive emotions			−0.323	7.720 ***
	Engagement			0.173	5.597 ***
	Relationship			0.067	1.912
	Meaning in life			−0.088	2.023 *
	Achievement			−0.027	0.661

Notes: Covariances include the year of study and average mobile phone use time per day on workdays and weekends; * *p* < 0.05, *** *p* < 0.001.

**Table 4 behavsci-13-00764-t004:** Mediating effect of PERMA wellbeing variables on the relationship between insomnia and psychological distress.

Effect	Effect Size	Boost SE	Boost 95% CI	
			BootLLCI	BootULCI
Total effects	1.053	0.028	0.998	1.108
Direct effects	0.942	0.028	0.887	0.997
Total mediation effects	0.111	0.015	0.083	0.144
Insomnia→Positive emotions→Psychological distress	0.137	0.024	0.094	0.188
Insomnia→Engagement→Psychological distress	−0.033	0.010	−0.054	−0.017
Insomnia→Relationship→Psychological distress	−0.027	0.015	−0.057	0.002
Insomnia→Meaning in life→Psychological distress	0.027	0.014	0.001	0.055
Insomnia→Achievement→Psychological distress	0.007	0.011	−0.014	0.029

Notes: n = 1714; bootstrap = 5000; → = unidirectional path; models were adjusted for year of study and average mobile phone use time per day on workdays and weekends.

## Data Availability

The data used to support the research findings are available upon request from the corresponding author. The data are not publicly available due to ethical restrictions.

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
