# Peer review of "Mediating Role of PERMA Wellbeing in the Relationship between Insomnia and Psychological Distress among Nursing College Students"

_behavsci, 2023, doi:10.3390/bs13090764_

Round 1
Reviewer 1 Report
Hi Dear,
My comments are attachted. Please improve the manuscript.

No
Author Response
Thank you very much for you review. Please find the attached responses to your comments.

Reviewer 2 Report
Thank you very much for invitation to review this interesting manuscript.
It is well--written and add to the current knowledge. However I have few minor notes:
1. Abstract: DO NOT STATR SENTENCE WITH NUMBER (LINE 16. (1,741).
2. Introduction: it is not clear why you introduced mobile phone addiction in line 40.
3. methods:
a) any scoring system and cut-off points for PERMA?
b) need to present the validity and reliability of all 3 instruments in previous studies.
c) indicate why you used Kruskal-Wallis (line 154) not ANOVA?
Results:
a) section 3.3 should be italized
b) line 201, check if the number of beta=0.667 correct. I confused and thinking it could be 0.597.
c) in line 207; you may report the result of mediation in terms of t, and p (path C')
d) make sure the numbers in table 4 are consistent with numbers in the figure 1.
Discussion: Any more practical suggestions for programs that enhance factrors related to PERMA (around lines 268-294).
Author Response

(The authors gave the same response as above.)

Reviewer 3 Report
Dear authors,
This work presents a cross-sectional study that examines the associations between insomnia and distress. Although it needs to be said that this is cross-sectional study, so no causal relationships can be determined. As a result from that, in the abstract and further in the text, it would be incorrect to state that this study is examining an effect. Please revise the entire paper on these statements and correct the statements according to a cross-sectional study that examines the association and not the effect.
Abstract line 12: the indirect effects? è please change this as this is a cross-sectional study
Lines 115-118: please indicate the rationale for not including marital status. Was the use of tabacco, alcohol or drugs questioned? If not, please indicate the rationale for not including this.
Lines 173: it is stated that there were great differences between year of study, therefore it would be great to divide the analyses (represented in tables 2, 3 and 4 for the separate groups, year of study one, two and three)
Line 222: remains? Should that be ‘examines’?
Lines 266-268: can this be substantiated by references?
Lines 284: please add a discussion section on the association of tabacco use, (or vaping), alcohol or drug use. As these factors are linked to wellbeing and most often used when feeling distressed, it is surprising that this was not used in the analyses. Please also add this in the limitations
Lines 285-294: please also indicate how stress can be reduced as a practical implication for use in practice. Add with reference.
Final spell check suggested
Author Response

(The authors gave the same response as above.)
